# Development of Perceived Complex Problem-Solving Instrument in Domain of Complex Systems

**Morteza Nagahi** [1,*] , **Alieh Maddah** [2] , **Raed Jaradat** [1] and **Mohammad Mohammadi** [2]

1   Department of Industrial and Systems Engineering, Mississippi State University,
    Mississippi State, MS 39762, USA; jaradat@ise.msstate.edu
2   Department of Management, Humanities Faculty, Birjand Branch, Islamic Azad University,
    Birjand 97177-11111, Iran; alieh.maddah68@alumni.ut.ac.ir (A.M.); Mohammadi@iaubir.ac.ir (M.M.)
*   Correspondence: mn852@msstate.edu

**Abstract:** The ability to solve modern complex systems becomes a necessity of the 21st century. The purpose of this study is the development of an instrument that measures an individual's perception toward solving complex problems. Based on literature and definitions, an instrument with four stages named perceived complex problem-solving (PCPS) was designed through exploratory and confirmatory stages. The instrument is validated and scaled through different models, and the final model is discussed. After completing validation and scale development of the PCPS instrument, the final model of the PCPS instrument was introduced to resolve the gap in the literature. The final model of the PCPS instrument is able to find and quantify the degree of perception an individual holds in dealing with complex problems and can be utilized in different settings and environments. Further research about the relationship between Systems Thinking and CPS revealed individuals with a high level of systems thinking have a better understanding of the characteristics of complex problems and so better perception of CPS.

**Keywords:** complex problem-solving; systems thinking; systems thinking skills/preferences; perceived complex problem-solving instrument; complex systems; exploratory and confirmatory factor analysis; scale development; SEM

## 1. Introduction

Modern complex systems deal with more socio-technical dimensions and interact directly with the surrounding environment, and this interaction creates challenges and issues [1]. The management of this turbulent work environment mandates the need for a skillset that involves creativity, continuous learning, innovation, and collaboration. Complex problem-solving (CPS) skills have become a necessary competence in today's workforce [2] and attract job seekers. This is evident through different programs that emphasize finding better approaches and methods in solving complex-system problem domains, for example, in different programs such as The Program for International Student Assessment (PISA) [3,4], The Program for the International Assessment of Adult Competencies (PIAAC) [5], and the O*net job database (the U.S. Department of Labor's Occupational Information Network) [6]. PISA is an assessment of the Organization for Economic Co-operation and Development (OECD), which includes the assessment of students' problem-solving skills and direct assessment of life competencies that apply across different areas of the school curriculum. PIAAC is an international assessment of adult skills managed by the OECD, which is currently being implemented by 25 countries in Europe, the Americas, and Asia. Although CPS has received attention in the literature, still not clearly defined, and the continued divergence in the definitions and perspectives will muddle the field and slow the progress of developing methods that can be applied to different disciplines [7].

Within the 21st century, modern complex systems still confront challenges with a high level of integration, ambiguity, uncertainty, and interdependence between systems and their related elements, making blurred the lines between technical, social, political, managerial, and organizational considerations [8,9]. Ackoff [10] claimed that one of the approaches that help us to evaluate and understand the complexities and challenges of third-millennium organizations is a systemic approach or systemic attitude, and he stated, in dealing with complex systems problems, one should focus on the system as a whole, rather than on the parts. In system theory, problems are studied based on their conditions, requirements, and developments, as well as their contributing factors and their interrelationships, are examined, and appropriate solutions are provided. Therefore, systems thinking is necessary for a more comprehensive and systematic approach in dealing effectively with modern complex systems and their problems/challenges. The study of factors that strengthen CPS skills helps employers hire competent employees and invest in their training.

In the 2000s, there was a belief that systems thinking can be an answer to complex systems problems [11–13], and there is convergence around their definitions [14,15], This belief was translated later into action, where some studies appeared to show the significance of systems thinking in the domain of complex systems and recruiting employees [16]. However, what remained unanswered is the relationship between an individual's systems thinking (ST) and his/her general perception of different stages in the CPS process—that is a current gap in the literature.

To address the gap and to improve the body of knowledge, the aims of the study are (1) to develop and validate a new perceived complex problem-solving (PCPS) instrument and (2) to investigate the relationship between ST and CPS using the developed instrument. The intent of the study is also to compare the effect of seven different dimensions of systems thinking, discussed later [1], on the performance of CPS.

The contribution of the study has three dimensions. From a methodological dimension, because of the simulation method in this field and the lack of an instrument that is easy to use in general and being base on CPS theories, this study develops and validates a new CPS instrument in the literature. Several validity and reliability measures are conducted to establish the development of the instrument. From a theoretical dimension, this study is important for academics since it helps to bridge the literature gap in the field by providing comparisons and relationships between different systems thinking dimensions with the perception of CPS stages. From a practical dimension, this study emphasizes on the importance of employees who obtain high-level ST and CPS skills to deal with modern complex system problems, so this study encourages HRM professionals to consider ST and CPS skills as work requirements in recruiting employees and hold training programs for both experienced managers and newcomers in the organization. This study can also be implemented in educational programs for students to evaluate and screen their skillset and capability in modern complex system problems.

An overview of CPS and ST is provided next, followed by the research hypotheses, the research methods, and the analyses performed to assess the validity and reliability of the theoretical model. The study concludes with a discussion, implications, limitations, and future research.

## 2. Background and Hypotheses

### 2.1. Complex Problem-Solving

### 2.1.1. Theories in Problem-Solving

In the way of solving a problem, there are different theories, which can be grouped into three.

(1)　Behaviorist: The behaviorists emphasize the role of stimulus–response interactions in problem-solving because of their emphasis on trial-and-error learning and habit strength [17].

(2)　Cognitive: By progression in cognitive psychology by the work of the Gestaltists, more focus and attention were devoted to the mental processes of learning and

problem-solving. This attitude believes that solving a problem is done in several steps and each step has a specific goal. Wallas and Polya were two cognitive psychologists who identified four stages of problem-solving [18,19]. The four stages of Wallas were preparation, incubation, inspiration, and verification, and the four stages of Polya were (a) problem understanding (b) plan devising; (c) carrying out the plan, and (d) backward-looking. Polya promoted general problem-solving strategies called heuristics and introduced them as keys in problem-solving expertise and intellectual performance. Heuristics methods were related to concepts such an analogical problem solving [20,21], symbolic problem solving [22,23], and abstract thought such as categorization, induction, and generalization [24–26].

(3) Information processing: Through further emergence of differences between experts and novices in solving problems, the information-processing theory of learning evolved. These theories pay attention to the exclusive nature of human beings in collecting and processing information to solve their problems, like computers to solve very complex problems. Base on this theory Newel and Simon work in artificial intelligence and combine thinking in human problem solving and artificial intelligence [27]. In this theory, problem solving is related to concepts such as working memory capacity [28,29], reasoning [30,31], long-term memory, and cognitive retrieval of relevant information [32].

Today, the dominant attitude in the study of problem-solving is the theory of information processing, which, like Gestalt's view of problem-solving, has certain assumptions. This theory believes that solving a problem is done in several steps and each step has a specific goal. Problem-solving is combined with evaluating the situation of a certain problem, selecting and applying the next specific steps, and this method continues until the end of problem-solving, just as a computer does.

### 2.1.2. Complex Problem-Solving

Modern complex problems are considered ill-defined problems with a lack of clear paths to obtain an optimal solution [33]. With the growth of complexity, it is difficult for problem solvers to evaluate the performance of the system since extracting information might be difficult to achieve. Therefore, the problem solver has interactions with the task until he/she gets information about progression [34] and reduces the gap between the initial state and the goal state by performing non-routine cognitive activities [14,35].

The research area in problem-solving began in cognitive psychology with the experimental work of the Gestaltists in Germany (e.g., Dunker, 1935 in [34]), typically with simple laboratory work (e.g., the "disk problem", later known as the "Tower of Hanoi") (e.g., [36] and Dunker's "X-ray" problem [37]), and it was thought it could be generalizable to more complex problems. At the beginning of the 1970s, researchers gradually became convinced that the theoretical concepts and empirical findings from simple laboratory tasks could not be generalized to complex real-world problems, and even under different circumstances, the basic CPS processes were different [38]. Since 1975, after global events such as the oil crisis, a new path has opened in the psychology of thinking that addresses complex problems and led to different reactions in North America and Europe [34]. The two ideas formed do not define problem-solving in the same way, and their divergent definitions led to different measurements of CPS.

(a) Two major approaches emerged in Europe, in Britain by Donald Broadbent [39], and in Germany by Dietrich Dörner [40,41]. Both approaches focused on complex laboratory tasks based on computer simulation, but these approaches differed somewhat in theoretical objectives and methods. In the British approach, mathematical problems were used in computer simulation systems to examine cognitive problem-solving processes under consciousness and unconsciousness.

In the German school, Funke and Frensch [34] stated that one obstacle must be removed in simple problem-solving, while in CPS, several obstacles require a set of cognitions

and prioritization programs to move forward the target situation. Dörner and Funke [42] claimed Funke and Frensch's definitions did not fully include the content or the relationship between the simulation and the real world. Therefore, they redefined a practical CPS as a collection of self-regulated psychological processes and activities that combine cognitive motivational and emotional aspects in a dynamic environment to achieve a bricolage and not perfect or optimal solutions. Complex problems require high knowledge and collaboration among many people [42]. In PISA 2012, the definition of CPS is the individual's capacity for cognitive processing to understand and solve problem situations [3]. The PISA 2015 defines collaborative problem-solving, and it showed that the students with a high level of collaborative problem-solving abilities could successfully carry out complicated problem-solving tasks with high collaboration complexity [4]. In PIAAC, it defines problem-solving in technology-rich environments [5].

Base on the German school definition, in the early 1980s, Dörner introduced the computer simulation scenario of "microworlds" such as Tailorshop [43] and "Lohhausen" [44], with several variables to allow experimental research of complex problems under controlled conditions [45]. Researchers in this field have found that, although the upper limit of complexity is not limited, the lower limits can be identifiable [46]. Therefore, they introduced "minimal complex systems" scenarios that consist of a single task or problem [47]. Then, a "multiple complex systems" approach [48] was introduced in response to the weaknesses of minimal complex systems.

(b)　The CPS definition in the North American approach emphasizes "the study of cognition in complex real-world conditions" [49] (p.135) and several techniques and tools developed in this approach. The O*net staff survey, which is the result of the efforts of the US Department of Labor, has developed several tools for measuring skills, knowledge, and abilities. It has assessed the importance of complex problems-solving in different occupations by eight items in the prototype version, then revised them in one item [6,50]. Although other tools such as personal problem-solving [51], managerial problem-solving [52], problem-solving styles [53], and social problem-solving [54] developed in this approach, still research for the development of a general theory in the evaluation of CPS abilities is not presented in the North American literature.

Despite much research in this area, the difference between the concept of a "simple problem" and a "complex problem" is still somewhat obscure, but we know that the greater the number of variables and the greater the relationships between them, cause the greater complexity of the problem [49,55]. It is still an open question which measurement can best assess the CPS or whether various other constructions should be proposed [7]. After an extensive survey in the works of literature that has attempted to expand the theories, definitions, and related concepts in the domain of problem-solving such as [18,19,27,31,33,36,42,43,51,56], the lack of a suitable questionnaire to assess recognition of CPS and its process is still a current gap. We used Stenberg's definitions in his book *Cognitive psychology* [33], about complex, insightful, and ill-structured problems and the processes of solving such problems, and also, the definitions and problem-solving processes in the prototype version of the O*net questionnaire and its revision [6,50], for designing an instrument to assess an individual's perception of CPS. The perceived problem-solving inventory does not directly assess problem-solving ability nor assesses one's function in a hypothetical problem situation. As stated in various sources in Heppner and Patersen [56], individuals act in hypothetical situations differently than real situations. This inventory evaluates a general knowledge of a person about complex problems and the process of solving them. True perception of CPS supports us in distinguishing it from simple problem-solving. Know that as barriers between a given state and a goal state are complex, change dynamically during problem-solving, and intransparent. Different aspects of a given state and the goal state are obscure for problem solvers and hard to identify. Solutions are not immediately obvious and are a combination of activities as a result of interaction between different solvers and their situation and are not necessarily perfect or optimal. Awareness

of these facts helps us to perform better and more realistically in passing the stages of real-world CPS.

In research conducted annually by The National Association of Colleges and Employers, problem-solving ability is one of the most important skills that employers seek on candidates' resumes. For example, the results of this annual survey showed that in 2016, employers, after the ability of the work team, are looking for problem-solving skills in work applicants [57]. This skill topped the list in 2017 [58], and in 2020 [59], respondents, with 91.2%, stated that it was the first skill they were looking for in a candidate's resume. Additionally, Mourshed and her colleagues [60], in their survey, stated that employers are looking for students with high problem-solving skills in the entry stage. In another study [61], it was shown that problem-solving skills lead to job success in new workforce entrants. In annual O*net surveys, the results show that problem sensitivity was among the top 10 job needs among the various occupations, and the greatest need for CPS is in occupations with the highest demands, financial values, and high rewards, such as senior executives, lawyers, judges, crisis management managers, or surgeons [6].

### 2.2. Systems Thinking

Numerous studies have linked complex systems and issues to systems thinking (ST) (e.g., [1,16,62–65]). Several researchers [14,15] stated that the definitions of CPS and ST have some overlap. Funke [14] stated that five attributes distinguish complex problems from simple problems, which include (1) the complexity of the problem situation, (2) the relationships between the variables involved, (3) the dynamics of the situation and developments within the system, and the role of time, (4) partial or complete lack of transparency, and (5) polytely (a Greek term for "many goals") and the possibility of conflict in the existence of several goals. Dörner and Funke [42] considered at least three aspects for complex systems: (1) Different levels of abstraction, (2) change (potentially unpredictable) over time, and (3) knowledge-rich with many potential strategies. Jaradat [1] introduced the characteristics of complex systems as (1) increasing complexity, (2) ambiguity, (3) high levels of uncertainty, (4) emergence, (5) evolutionary development, (6) interconnectivity, and (7) integration.

According to Checkland [66], ST is the ability to think and speak in a holistic language to understand and deal with complex system problems. Flood and Carson [67] and Richmond [68] define ST as a framework that helps individuals to address complex things. Jaradat and his colleagues stated that an individual's systemic thinking capacity could be an effective response to a complex system problem [1,9]. Although some tools and techniques have been developed for ST such as [69,70], Jaradat and his colleagues developed a systems thinking skills preferences (STSP) instrument (with $\alpha = 0.91$) based on the grounded theory method, which is the first instrument for evaluating an individual's systemic thinking capacity, it includes seven dimensions: (1) level of complexity, (2) level of independence (autonomy), (3) level of interaction, (4) level of change, (5) level of uncertainty, (6) level of the systems worldview (hierarchical view), and (7) level of flexibility (see Figure 1) [1,9]. This instrument was used in data collection for obtaining participants' predisposition for ST skills.

### 2.3. Hypotheses Development and the Proposed Theoretical Model

In research, ST has been conceptualized in relation to dealing with complex systems and problems. However, there are still gaps in this area.

a.  Although Maani and Maharaj [13] have attempted to show the relationship between ST and performance in CPS in a sample of 10 participants, the relationship between ST and the general perception of complex problems nontransparent aspects without specific training in CPS has not yet been investigated.

b.  Most of the complex problems-solving research belongs to the German school and are based on computer simulation. In the North American approach, questionnaires were developed in the field of CPS importance [6], personal problem-solving [71], problem-

solving styles [53], and social problem-solving [54], regardless of novelty, simplicity, or complexity of problems, and whether or not there are single or multiple barriers or goals. Therefore, there is a lack of questionnaires that assess perceived complex problems-solving based on theories and are easy to use for students, administrators, and employees.

**Less Systemic (Reductionist)** | **Dimension** | **More Systemic (Holistic)**

**Simplicity (S):** Avoid uncertainty, work on linear problems, prefer the best solution, and prefer small-scale problems.

**Level of Complexity:** Comfort with multidimensional problems and limited system understanding.

**Complexity (C):** Expect uncertainty, work on multidimensional problems, prefer a working solution, and explore the surrounding environment.

**Autonomy (A):** Preserve local autonomy, a trend more toward an independent decision and local performance level.

**Level of Independence:** Balance between local level autonomy versus system integration.

**Integration (G):** Preserve global integration, a trend more toward dependent decisions and global performance.

**Isolation (N):** Inclined to local interaction, follow a detailed plan, prefer to work individually, enjoy working in small systems, and interested more in causeeffect solution.

**Level of Interaction:** Interconnectedness in coordination and communication among multiple systems.

**Interconnectivity (I):** Inclined to global interactions, follow a general plan, work within a team, and interested less in identifiable cause-effect relationships

**Resistance to Change (V):** Prefer taking few perspectives into consideration, over specify requirements, focus more on internal forces, like short-range plans, tend to settle things, and work best in a stable environment.

**Level of Change:** Comfort with rapidly shifting systems and situations.

**Tolerant of Change (Y):** Prefer taking multiple perspectives into consideration, underspecify requirements, focus more on external forces, like long-range plans, keep options open, and work best in a changing environment.

**Stability (T):** Prepare detailed plans beforehand, focus on the details, uncomfortable with uncertainty, believe the work environment is under control, and enjoy objectivity and technical problems.

**Level of Uncertainty:** Acceptance of unpredictable situations with limited control.

**Emergence (E):** React to situations as they occur, focus on the whole, comfortable with uncertainty, believe the work environment is difficult to control, and enjoy nontechnical problems.

**Reductionism (R):** Focus on particulars and prefer analyzing the parts for better performance

**Systems Worldview:** Understanding system behavior at the whole versus part level.

**Holism (H):** Focus on the whole, interested more in the big picture, and interested in concepts and abstract meaning of ideas.

**Rigidity (D):** Prefer not to change, like determined plans, not open to new ideas, and motivated by routine.

**Level of Flexibility:** Accommodation of change or modifications in systems or approach.

**Flexibility (F):** Accommodating to change, like a flexible plan, open to new ideas, and unmotivated by routine.

**Figure 1.** Seven dimensions of the "ST Skills Preferences Instrument" [1].

In this study, to address these gaps, a questionnaire was developed to assess the individual's perceptions of CPS, and its validity and reliability evaluated by factor-analysis

results, in addition to providing an examination of the relationship between systems thinking and perceived complex problem-solving, which enriches the body of current literature.

*2.4. The Relationship between Systems Thinking and Complex Problem-Solving*

In many studies, ST is considered an appropriate response to complexity because it provides a more holistic view of a problem area [9]. Senge [72] argued that due to overwhelming complexity, ST is needed more than ever. Richmond [73] described ST as a superior approach in dealing with complexity. Sweeney and Sterman developed a list of systemic thinking features to assess students' capability in complexity [62]. In another study, Keating, Kaufman, and Dreyer examined whether ST in an organization could provide a framework for analyzing and solving complex issues. The results of this study showed that ST can prepare us to solve problems effectively in today's turbulent environment and can be used as a suitable framework for analyzing and solving problems in the management of organizations [12]. Jackson [11], in his study on the effectiveness of the use of ST in solving complex social problems, showed that ST could be used as a coherent method to solve social problems. In another study in the information and communications technologies sector, Petkov and his colleagues [74] showed that techniques from soft systems and multiple criteria decision making (MCDM) could be effective in particular stages of a CPS intervention. Considering the widespread belief about the connection between ST and complexity, Mani and Maharaj [13] examined the relationship between ST and performance in CPS for empirical substantiation of this belief. Based on simulation tests, they showed a certain type of ST, and more importantly, the subject's approach to the problem is relevant to solving a problem.

Due to the five features of the complex problem [14,49] and the features of complex systems [1,9,42] (as described in the previous section) and the ST skills [1], it is evident that many of the CPS can be managed through ST. ST skills help individuals understand the structure of problems, leading to better performance in problem-solving in complexity [13] (p. 7). However, overall, what remains neglected in studies is the effect of ST on the general perception of complex problems and their nontransparent aspects. Therefore, in this study, this issue has been considered, and different skills of ST on PCPS are evaluated.

## 3. Methodology

In this study, after validation of the perceived complex problem-solving (PCPS) instrument, the relationship between ST and PCPS was examined. In other words, we investigated the impact of systems thinking skills preferences (STSP) on the PCPS of managers and students. To measure this relationship, two studies were performed. The first study targeted managers who face high levels of complex system problems in their organizations, and the second study targeted students as prospective future workforce. Two different samples were considered for testing the construct validity and internal consistency of the theoretical model across different samples. Figure 2 shows the research framework.

*3.1. Materials*

In this study, two questionnaires were used: The Systems Thinking Skills preferences (STSP) Questionnaire (with $\alpha = 0.91$), developed by [1,9], with 39 questions, evaluates seven preferential categories/systems skills dimensions (Figure 1) and determines the individual's desire for Holistic or Reductionist thinking. Based on these dimensions, one score determines the total ST score for each individual. Due to the lack of a suitable questionnaire to assess CPS abilities, a questionnaire consisting of nineteen five-point Likert scale questions is developed and tested for validity and reliability (with 0.89). The questionnaire consists of four stages of CPS: (1) Problem Identification and Definition (questions 1–5; an example question in this dimension designed for students is "I am often facing unique and new problems in my engineering coursework."), (2) Information Gathering about problems and solutions (questions 6–11; an example question designed for students is "The methods, resources, or people through which information can be

collected are not recognized well."), (3) Evaluating solutions and Developing Approaches (questions 12–16; an example question in this dimension designed for students is "It is hard to evaluate and assess the strengths and weaknesses of new ideas and solutions."), and (4) Implementation Planning (questions 17–19; an example question in this dimension designed for students is "It is difficult to present and develop an executive plan for the realization of new ideas."), which totally assesses the PCPS (see Appendix A. Table A1). All items are scored on a five-point Likert scale, ranging from 1 = strongly disagree to 5 = strongly agree. A total score can be calculated as a general index of the PCPS of a person.

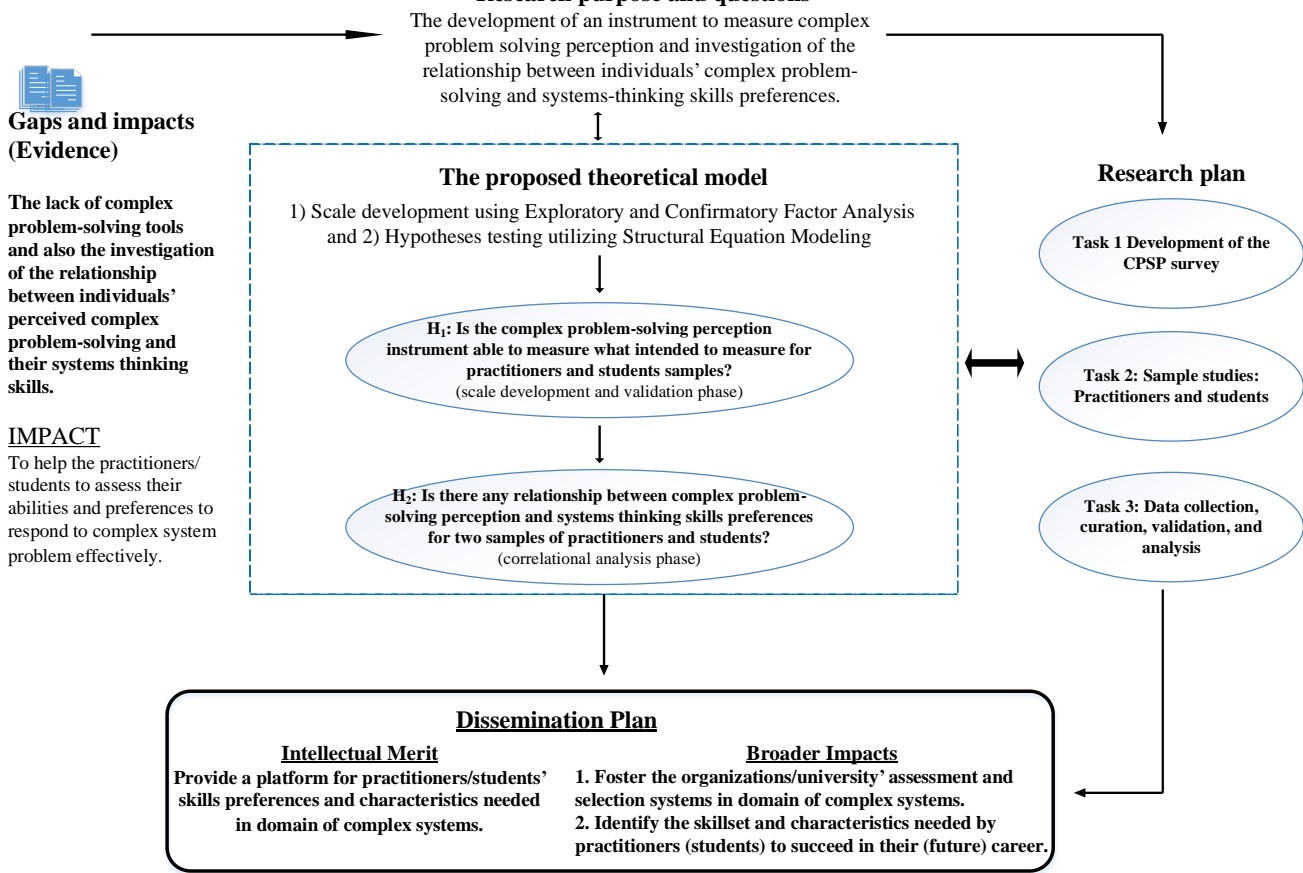

**Figure 2.** Research framework.

These questionnaires are used to measure individuals' assessment of their perception to CPS and determine their ST skills. Demographic factors are added to the proposed theoretical model.

*3.2. Sample and Data Collection Procedure*

3.2.1. Study 1

Participants

The statistical population of this study was managers of the governmental executive organizations in the South Khorasan Province in Iran. The respondents were n = 250, including 49 females and 201 males, and three CEOs, 46 deputies, and 201 office managers. Respondents answered questions related to their age, managerial background, and work experience. The sample characteristics are shown in Table 1.

**Table 1.** Sample characteristics (Study 1).

| Variable | Categories | Number (Percentage) |
|---|---|---|
| Gender | Male | 80.4% |
| | Female | 19.6% |
| Age | ≤30 | 1.6% |
| | 31–40 | 36.4% |
| | 41–50 | 50.0% |
| | 51–60 | 10.8% |
| | ≥60 | 1.2% |
| Level of education | High school diploma | 0.0% |
| | Bachelor's degree | 31.2% |
| | Master's degree | 56.0% |
| | Ph.D. | 12.8% |
| The major of study in the highest degree | Engineering | 39.2% |
| | Social science | 14.8% |
| | Business/Management | 28.0% |
| | Health-related | 2.0% |
| | Others | 16.0% |
| Work experience (year) | Less than 10 | 8.8% |
| | 11–20 | 48.4% |
| | 21–30 | 36.4% |
| | More than 30 | 6.4% |
| Management experience (year) | Less than 10 | 58.8% |
| | 11–20 | 33.6% |
| | 21–30 | 6.4% |
| | More than 30 | 1.2% |
| Managerial level | CEO | 1.2% |
| | Vice president/Deputy | 18.4% |
| | Office manager | 80.4% |

Procedure

Step 1. The development of a perceived complex problem-solving questionnaire

The initial version of the questionnaire was developed to assess an individual's perception of CPS. In order to determine its validity and reliability, according to [75], the initial version of the PCPS questionnaire was given to 10 experts teaching public administration and management at different universities. The validity of its content (the relevance of the phrase, simplicity of the phrase, and the clarity of the phrase) was evaluated. Questions were accepted with CVI > 0.7, and then its reliability was evaluated among 250 employees with $\alpha = 0.895$. All "Cronbach's Alpha if Item Deleted" values were less than the overall Cronbach's Alpha of 0.895, suggesting all questions are reliable.

Step 2. The translation of the STSP Questionnaire

According to the literature [76,77], the STSP questionnaire were translated from their original form into the Persian language. The STSP instrument is translated to the Persian language through a panel of experts to better accommodate the language used by participants and to obtain a valid analysis. Then, by comparing the two versions, modifications were made. The instrument was given to a small group of managers, and the reliability was evaluated with $\alpha = 0.841$, and the final survey was produced. All "Cronbach's Alpha

if Item Deleted" values were less than the overall Cronbach's Alpha of 0.841, suggesting all questions are reliable.

The Persian version of the PCPS and ST questionnaires was used in this study. The sample size consisted of seventeen governmental executive organizations of South Khorasan. The selection criteria were based on stratified random sampling. Four hundred-fifty paper questionnaires were distributed among CEOs, deputies, and office managers of provincial organizations in the summer of 2020, and 250 questionnaires were returned.

3.2.2. Study 2
Participants

The statistical population of this study was students at Mississippi State University in the United States. Four hundred eighty-one students participated in the study. From 481 collected responses, 373 students' responses were analyzed. The pair-wise deletion has been used in data analysis. The sample characteristics are shown in Table 2. The percentage of female and male respondents were 35.9% and 64.1%, respectively, and 67.3% undergraduate and 32.7% graduate students. Their age range was from 18 to 60 with a mean of 28.7 years and SD of 10.0 years, and they were made up of 83.9% full-time students and 16.1% part-time students. Additionally, 9.9% were distance learning students, and 90.1% were on campus. The mean CGPA of students was 3.45, with an SD of 0.54 ranging from 2.00 to 4.00. They have passed an average of 54.6 credits/hours in their program with an SD of 37.6.

**Table 2.** Sample characteristics (Study 2).

| Variable | Categories | Number (Percentage) |
|---|---|---|
| Gender | Male | 63.8% |
| | Female | 36.2% |
| Ethnicity and Race | Asian | 12.3% |
| | African American | 5.0% |
| | Caucasian | 72.7% |
| | Hispanic | 2.3% |
| | Middle Eastern | 2.3% |
| | Multi-racial | 3.1% |
| | Native American | 1.2% |
| | Prefer not to disclose | 1.2% |
| Currently employed (not including co-op/internship) | No | 54.2% |
| | Yes | 45.8% |
| Completed a co-op | No | 83.1% |
| | Yes | 16.9% |
| Completed a professional internship | No | 78.1% |
| | Yes | 21.9% |

Procedure

A web-based survey was used to collect data for this study, and emails were sent to students in the Fall of 2020–2021. In this study, the original version of the STSP instrument [9] and the English version of the PCPS instrument were used.

## 4. Data Analysis
### 4.1. Factor Analysis and Scale Development

The purpose of this study is to bridge the literary gap with regard to an instrument for defining the PCPS of an individual. To meet this end, an individual's perception will be

analyzed when faced with modern complex system problems. The scale development was conducted in two main stages—-the exploratory and confirmatory stage. Other studies have applied similar development framework scales, initiated by studies with the pilot test (gathering experts' feedbacks), followed by a meticulous construction of the validity in EFA (exploratory stage). Finally, the framework is completed by constructing validity analysis using CFA (confirmatory stage) [78–81].

In the exploratory stage, to achieve an initial theoretical model of the PCPS instrument, the KMO test, Bartlett's test of Sphericity, and anti-image correlation matrix have been done to assure the sample size is adequate. Then, EFA framework performed and showed four factors retained with eigenvalues greater than one. The reliability for both studies were more than 0.8, which were very good. After EFA procedures, the baseline model of the PCPS instrument with four main factors/constructs and 19 items has been designed. In the next stage, a confirmatory factor analysis (CFA) procedure was done, and by measuring construct validity, uni-dimensionality, discriminant validity, and composite reliability (CR), the final structural model of the PCPS instrument was provided. Finally it has proved that the final model of the PCPS instrument fits the data well and is able to measure the state of PCPS at the individual level. For more details about the analysis procedures and development of the instrument, see Appendix B.

*4.2. Structural Equation Modeling (SEM)*

Study Variables

The variables listed below are developed in the proposed theoretical model (see Figure 3).

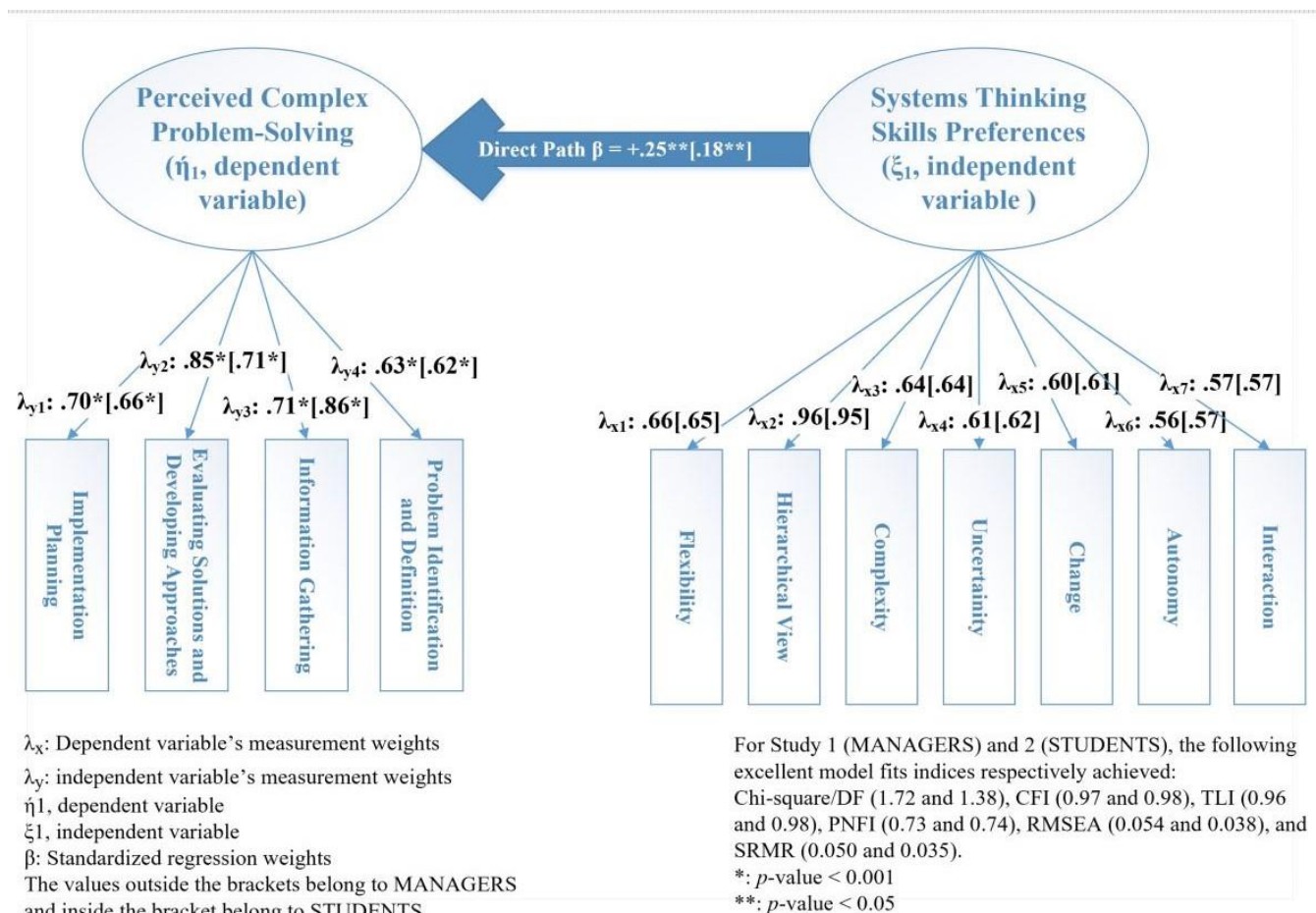

**Figure 3.** The full structural model analysis of the proposed theoretical model for both samples of practitioners and students.

Latent Independent Variable

The "Systems Thinking Skills Preferences (STSP)" is an abstract theoretical variable and cannot be directly measured; therefore, we used a latent variable (unobservable variable) to indirectly measure it through the seven observed variables associated with the seven dimensions of the STSP instrument. This latent variable indirectly measures the individuals' overall systemic skills preferences based on the seven dimensions, which resulted from an extensive systematic review using grounded theory in the domain of complex systems. The seven dimensions are (1) level of Complexity, (2) level of Independence, (3) level of Interaction, (4) level of Change, (5) level of Uncertainty, (6) level of Systems Worldview, and (7) level of Flexibility. Figure 1 indicates the detailed definition of each dimension with a simple description of each.

Latent Dependent Variable

To assess individuals' PCPS, the study utilized the PCPS instrument with its four stages: (1) Level of Problem Identification and Definition, (2) Level of Information Gathering, (3) Level of Evaluating Solutions and Developing Approaches, and (4) Level of Implementation Planning dimensions. These four dimensions, which are condensed into one latent variable called PCPS, are used as a problem-solving perception indicator for the study's population.

Before interpreting the results of the study, the proposed theoretical model needs to be validated through the establishment of construct validity. As mentioned, the proposed theoretical model shows the structural relationship between dependent and independent latent variables (that is, STSP and PCPS) through the regression and measurement weights.

The construct validity of the theoretical model is obtained through the investigation of model fit indices. The fit indices values indicated that the proposed theoretical model obtained the construct validity and measured what it is intended to measure; consequently, it is deemed valid to test the study's hypotheses. The construct validity was conducted (1) to show that the proposed theoretical model was able to measure what it is intended to measure (i.e., the proposed model fits the data), (2) to show that the associated results of the model can be generalizable, and (3) to test the study hypotheses.

To test the study hypotheses, the proposed theoretical model was tested through structural equation modeling using AMOS software version 25.0. The standardized solution for the theoretical model consists of the full structural model used to assess all the relationships among the study's variables (see Figure 3).

As seen in Figure 3, practitioners/students with high scores on the ST dimensions of levels of Complexity, Independence, Interaction, Change, Uncertainty, Systems Worldview, and Flexibility also have high scores on four stages of PCPS, including (1) Level of Problem Identification and Definition, (2) Level of Information Gathering, (3) Level of Evaluating Solutions and Developing Approaches, and (4) Level of Implementation Planning dimensions. For example, a practitioner/student with a high score in the Level of Problem Identification and Definition dimension indicates his/her better understanding and defining the problems, and a practitioner/student with a high score in the *Complexity* dimension indicates his/her clear skill preference toward Complexity compared to Simplicity (see Figure 1). The practitioners with low scores on the seven dimensions of ST skills preferences are associated with low scores on the four stages of PCPS.

Since the relationship between the ST Skills Preference and PCPS latent variables is significant with *p*-value < 0.001 (*t*-value = 3.31) and standardized regression weight of $\beta_1$ = +0.25 (with the standard error of 0.03) for practitioners in study 1 and with *p*-value of 0.013 (*t*-value = 2.47) and standardized regression weight of $\beta_1$ = +0.18 (with the standard error of 0.003) for students in study 2, the main hypothesis is supported. This indicates that the ST skills preferences of practitioners/students have a positive relationship with their PCPS. In other words, the ST of practitioners/students affects their PCPS.

## 5. Discussion

The competitive environment, rapid changes, and the expansion of communication have led organizations to complex systems with multiple relationships. In such situations, complex challenges and problems have arisen, and as a result, the ability to solve complex problems is a necessary competency for an individual and organization. Therefore, complex problem-solving (CPS) has been considered in numerous international evaluations both in the field of education and in the industry.

In Phase I of the study, the literature about the history, definitions, and process of CPS were reviewed. Most assessments of CPS were using computer simulations, and there was no questionnaire for professional assessment with regard to other questionnaires like personality, critical thinking, and performance. Although several typical problem-solving questionnaires were designed in specific areas regardless of the simplicity or complexity of the problem, a questionnaire based on CPS definitions does not exist. As a result, to bridge this literature gap, a questionnaire was designed in Phase II. In this phase, based on theories and processes, four main stages were derived, and 32 phrases were designed for the purpose of assessing the level of general knowledge and understanding of people about complex problems and the processes needed to solve them. Then, in Phase III, after gathering experts' feedback and ideas, 19 items were chosen, and the PCPS instrument was developed. The content validity of the questionnaire (the relevance of the item, simplicity of the item, and the clarity of the item) was evaluated by ten university faculties and experts, and all 19 questions were accepted with CVI > 0.7. The main purpose of this phase was to determine the capability of the instrument to capture an individual's PCPS.

Along with using the PCPS instrument to gather data, the scale development of the instrument was started in Phase IV. In the data collection of two studies, 250 managers and 373 students from different races, gender, educational backgrounds, and occupations have participated in the experiment. This dataset had no missing value and passed normality test criteria. Some comprehensive scale development techniques were performed in two stages called the exploratory stage and the confirmatory stage. To shape the initial theoretical model, the dataset has been analyzed in the exploratory factor analysis framework and resulted in the initial theoretical model called the baseline model. To make the final decision about the number of factors, after checking eigenvalues and the scree plot, four factors were retained with eigenvalues greater than one, including Level of Problem Identification and Definition, Level of Information Gathering, Level of Evaluating Solutions and Developing Approaches, and Level of Implementation Planning.

After attaining the initial theory of the PCPS instrument, the confirmatory stage began to test the initial theoretical model. In the confirmatory stage, the baseline model was tested and modified through the CFA framework. After completing six main steps of CFA, the best-fitted model to the dataset called the final model was retained. The final model consisted of four distinct factors (constructs) and 17 items (questions), which measure different individuals' PCPS. The final model had the best theoretical and logical support along with good construct validity and reliability results, and it will service as the validated theoretical model for the PCPS instrument and measure the level of perception of individuals in CPS.

The PCPS instrument presented in this study allows for better understanding with regard to individuals' PCPS. The application of this instrument is broad with usefulness in industry, education, and government, and will allow management/superiors to identify the strengths and weaknesses of an individual in terms of cognitive thinking. Therefore, for further research in this study, the tool has been used to assess the relationship between an individual's systems thinking skills preferences (STSP) and his/her PCPS. Base on testing, the main hypothesis is supported. This indicates that the STSP of practitioners/students have a positive relationship with their PCPS. In other words, practitioners/students with high scores on ST dimensions of levels of Complexity, Independence, Interaction, Change, Uncertainty, Systems Worldview, and Flexibility also have high scores on four stages of PCPS, including (1) Level of Problem Identification and Definition, (2) Level of Information

Gathering, (3) Level of Evaluating Solutions and Developing Approaches, and (4) Level of Implementation Planning dimensions. The contribution of this hypothesis is consistent with other studies such as [62], who developed a list of ST features to assess students' capability in complexity. Keating and his colleagues [12] showed ST could provide a framework for analyzing and solving complex issues in the management of today's organizations. Mani and Maharaj [13] showed ST has a relationship with performance in CPS. As they mentioned, ST aids in understanding the structure of a problem and then would lead to better performance.

*Future Studies and Limitations*

This tool does not directly assess problem-solving ability, but rather examines the level of perception of individuals from complex problems and complex problem-solving processes. The higher a person's score in PCPS, the better their knowledge and understanding of CPS and its process for achieving more effective results. This test does not ask the participants about a hypothetical and specific situation and neither designed for a specific setting like management or education, etc., so it can be used in different settings wherever individual needs to deal with complex problems. For this goal, further research by investigating many ways of applying the tool in a more interactive setting and comparing new and old results for improving the reliability of the instrument further.

## 6. Conclusions

The intent of this study was to test the validity of the PCPS instrument in the domain of complex systems, so two studies were done segregated, and the results were reported separately. As the two sample populations were in different settings, cultures, and ages, they were not integrated. The results of the two studies showed this instrument can be used in other populations with different settings and can probably remain valid and reliable. In addition, this investigation gave an assessment of the connection between systems thinking (ST) and perceived complex problem-solving (CPS), which advances the assortment of the current literature.

**Author Contributions:** Conceptualization, M.N., A.M. and M.M.; methodology, M.N., A.M. and M.M.; software, M.N.; validation, M.N., A.M., R.J. and M.M.; formal analysis, M.N.; investigation, M.N., A.M. and M.M.; resources, M.N., A.M. and M.M.; data curation, M.N. and A.M.; writing—original draft preparation, M.N. and A.M.; writing—review and editing, M.N., A.M. and R.J.; visualization, M.N. and A.M.; supervision, M.N., R.J. and M.M.; project administration, M.N. and A.M.; funding acquisition, R.J. and A.M. All authors have read and agreed to the published version of the manuscript.

**Funding:** This research received no external funding.

**Institutional Review Board Statement:** The study was conducted according to the guidelines of the MSU HRPP, and approved by the Institutional Review Board of Mississippi State University (protocol code IRB-19-401 and 25 November 2019).

**Informed Consent Statement:** Informed consent was obtained from all subjects involved in the study.

**Conflicts of Interest:** The authors declare no conflict of interest.

## Appendix A

**Table A1.** Perceived complex problem-solving questionnaires.

| Problem Identification and Definition | Identification: Identifying the Nature of Problems and the Goal We Want to Achieve. (Find Out What the Problem Is?) Definition: What Information Does the Problem Give Us and What Does It Ask? And Redefine the Problem | | | | |
|---|---|---|---|---|---|
| 1. It is difficult to accurately identify and define problems and issues related to my coursework in the major of engineering study [my job function and tasks] [33]. | Strongly disagree | Disagree | Neutral | Agree | strongly agree |
| 3. It is difficult to accurately identify the different dimensions and aspects of problems related to my coursework in the major of engineering study [my job function and tasks] [33]. | Strongly disagree | Disagree | Neutral | Agree | strongly agree |
| 4. I am often facing unique and new problems in my engineering coursework [my job function and tasks for managers]. | Strongly disagree | Disagree | Neutral | Agree | strongly agree |
| 5. Problems in my engineering coursework [my job function and tasks] are hard to predict and expect. | Strongly disagree | Disagree | Neutral | Agree | strongly agree |
| 6. Problems in my engineering coursework [my job function and tasks] are difficult to analyze. | Strongly disagree | Disagree | Neutral | Agree | strongly agree |
| information Gathering about problems and solutions | Knowing how to find information and identify essential information. | | | | |
| 7. It is hard to gather the information needed to make decisions in my field of engineering study [my job function and tasks]. | Strongly disagree | Disagree | Neutral | Agree | strongly agree |
| 8. It is often hard to identify the causes of the problems and issues that I face in my coursework in the major of engineering study [my job function and tasks] [33]. | Strongly disagree | Disagree | Neutral | Agree | strongly agree |
| 9. The methods, resources, or people through which information can be collected are not recognized well in my coursework in the major of engineering study [my job function and tasks]. | Strongly disagree | Disagree | Neutral | Agree | strongly agree |
| 10. In most cases, I feel that there is no expertise to gather information about the causes of problems in my coursework [my job function and tasks]. | Strongly disagree | Disagree | Neutral | Agree | strongly agree |
| 11. It is often hard to identify potential solutions to problems and issues well. | Strongly disagree | Disagree | Neutral | Agree | strongly agree |
| 12. It is hard to reorganize the information collected to identify new solutions in my coursework in the major of engineering study [my job function and tasks]. | Strongly disagree | Disagree | Neutral | Agree | strongly agree |

**Table A1.** *Cont.*

| Problem Identification and Definition | Identification: Identifying the Nature of Problems and the Goal We Want to Achieve. (Find Out What the Problem Is?) Definition: What Information Does the Problem Give Us and What Does It Ask? And Redefine the Problem | | | | |
|---|---|---|---|---|---|
| Evaluating solutions and Developing Approaches | Developing Approaches and Evaluating the likely success of an option in reaction to the demands of the situation. | | | | |
| 13. It is hard to classify the information obtained to evaluate potential solutions in my coursework in the major of engineering study [my job function and tasks]. | Strongly disagree | Disagree | Neutral | Agree | strongly agree |
| 14. It is often difficult to predict the potential outcomes of the solutions in my coursework in the major of engineering study [my job function and tasks]. | Strongly disagree | Disagree | Neutral | Agree | strongly agree |
| 15. The problems I face require new solutions and creative ideas in my coursework in the major of engineering study [my job function and tasks]. | Strongly disagree | Disagree | Neutral | Agree | strongly agree |
| 16. It is hard to evaluate and assess the strengths and weaknesses of new ideas and solutions in my coursework in the major of engineering study [my job function and tasks]. | Strongly disagree | Disagree | Neutral | Agree | strongly agree |
| 17. In addition to assess strengths and weaknesses of new ideas, the possibility of successful implementing is hard to be explored in my coursework in the major of engineering study [my job function and tasks]. | Strongly disagree | Disagree | Neutral | Agree | strongly agree |
| Implementation Planning | Developing approaches for implementing an idea. | | | | |
| 18. In the area of my coursework problems, beside coming up with new ideas and solutions, it is hard to expect presenting an executive plan to implement the new ideas and solutions in my coursework in the major of engineering study [my job function and tasks]. | Strongly disagree | Disagree | Neutral | Agree | strongly agree |
| 19. It is difficult to present and develop an executive plan for the realization of new ideas in my coursework in the major of engineering study [my job function and tasks]. | Strongly disagree | Disagree | Neutral | Agree | strongly agree |
| 20. There does not exist the required competencies and capabilities to develop an executive plan to implement the ideas in my major of engineering study [my job function and tasks]. | Strongly disagree | Disagree | Neutral | Agree | strongly agree |

The phrase "my coursework in the major of engineering study" was used in students' questionnaires, and the phrase "my job function and tasks" was used in managers' questionnaires.

**Appendix B**

*Appendix B.1. Factor Analysis and Scale Development*

Exploratory Factor Analysis (EFA) procedures were conducted as the dimension reduction (data-driven) technique using SPSS software, version 26; this shapes the initial theoretical model for the PCPS called the "baseline model" [82]. The CFA, unlike EFA, is a theory-driven technique that requires a priori theoretical model (priori for this study was the baseline model resulted in EFA). Confirmatory factor analysis (CFA) procedures acted as the confirmatory stage utilizing AMOS, version 25, to confirm the structure of the baseline model. The CFA provided several analytics, including theory and hypothesis testing through construct validity, evaluation of method effects, examination of the stability of the factor model over participants, and a correlation between error terms.

*Appendix B.2. Exploratory Stage*

In the exploratory stage, factor analysis using SPSS software to determine the initial number of latent factors and respective items for each latent factor (construct) for the PCPS instrument. The following steps were conducted in the exploratory stage to achieve an initial theoretical model of the PCPS instrument.

Appendix B.2.1. Sample Size Adequacy

The data should be appropriate for the use of factor analysis [83]. To assure sample size adequacy, three criteria have been tested, including the KMO test, Bartlett's test of Sphericity, and anti-image correlation matrix. The adequate results have been achieved from KMO (study 1: 0.89 > 0.50 and Study 2: 0.88 > 0.50) and Bartlett test (study 1: Chi-square (136) = 1821.4, $p < 0.001$ and study 2: Chi-square (171) = 1876.1, $p < 0.001$) [84,85]. In the anti-image correlation matrix, high inter-correlations depict the importance of an item to a factor [84]. The matrix showed that almost all of the items loaded higher than 0.40 in respective factors, and there was no extreme multicollinearity between the items. These results prove that the data and sample size are appropriate for factor analysis (EFA framework).

Appendix B.2.2. Exploratory Factor Analysis Procedure

To perform EFA framework, a decision should be made in four criteria: (1) factor extraction method, (2) factor rotation method, (3) factor selection, and (4) choosing association matrix. Principal components analysis is the most frequently used EFA extraction method [84] and has been chosen as the extraction method. To interpret the meaning of the four retained factors, orthogonal (Varimax) rotation has been chosen as the factor rotation method.

Factor Selection: To make the final decision about how many factors should be extracted, two criteria have been checked: (a) Eigenvalues shows variance explained by that particular factor out of the total variance [84]. Four factors have been kept with eigenvalues greater than one using Kaiser's criterion of retaining. (b) The aim of the scree plot is to determine the optimal extracted factors. All the factors on the steep slope should be retained, and the other factors should be neglected [84]. Using the scree plot, four factors retained with eigenvalues greater than one.

These four factors extracted in EFA measure the four stages of the PCPS instrument, including Level of Problem Identification and Definition, Level of Information Gathering, Level of Evaluating Solutions and Developing Approaches, and Level of Implementation Planning stages. Table A2 shows the factors' operational definitions and respective descriptions.

Reliability: Cronbach's Alpha is conducted and yielded very good results in studies 1 and 2 with 0.92 and 0.89, respectively (Alpha greater than 0.8 and 0.9 is very good and Excellent, respectively) [86].

**Table A2.** Factors and Respective Operational Definition.

| Construct | # of Qs | Description | Operational Definition |
|---|---|---|---|
| $\lambda_1$ | 5 | Items related to Problem Identification and Definition | Problem Identification: Identifying the nature of problems and the goal we want to achieve. (Find out what the problem is?) Problem Definition: What information does the problem give us, and what does it ask? And redefine the problem. |
| $\lambda_2$ | 6 | Items related to Information Gathering about problems and solutions | Information Gathering: Knowing how to find information and identify essential information. |
| $\lambda_3$ | 5 | Items related to Evaluating Solutions and Developing Approaches to problems | Evaluating Solutions and Developing Approaches: Developing Approaches and Evaluating the likely success of an option in reaction to the demands of the situation. |
| $\lambda_4$ | 3 | Items related to Implementation Planning for problems and solutions | Implementation Planning: Developing approaches to implementing an idea or solution. |

After completing the EFA procedures, the initial model of the PCPS instrument has been designed—the baseline model. The baseline model consisted of four main factors/constructs and 19 items with 19 corresponding loadings. This multi-vocal model served as the initial model to start CFA procedures. The confirmatory stage has been designed and conducted to test the initial theory from the exploratory stage and, if necessary, whether to correct the baseline model or conduct a new model. The next section provides a confirmatory framework along with a detailed illustration of the final structural model of the PCPS instrument.

*Appendix B.3. Confirmatory Stage*

Confirmatory Factor Analysis (CFA) is applied when researchers have clear hypotheses regarding a specific scale or instrument—-the baseline model from the exploratory stage. CFA can be used to test whether the items are related to the hypothesized latent constructs as expected, and also the model has a sufficient number of latent constructs. If the CFA test finds this relationship, then the model will achieve structural construct validity [87]. The inability of the exploratory stage to clearly explain relationships between items with their respective latent constructs makes EFA far less suitable for the purpose of scale development and construct validity [88]. As such, the CFA is found to be more powerful and appropriate for theory and scale development [88]. There are several beneficial software packages that may be used to conduct CFA; while any of the major software packages would work well, Amos 25.0 was selected for its ease of use and user interface.

Appendix B.3.1. Confirmatory Factor Analysis Procedure

The CFA application is comprised of six steps. It starts from model specification, followed by model identification, parameter estimation, the model fit, and finally, the end model is re-specified and compared with other rival models [89]. In this section, the six steps consecutively have been explained. (1) Model Specification is concerned with formulating a model based on a theory and/or previous studies in the field [87]. Initial relationships between variables need to be made clear. The initial theoretical model—-the baseline model obtained from the exploratory stage—was used in the confirmatory stage. (2) Model Identification is concerned with whether one can derive a unique value for each parameter

whose value is unknown [87]. The model was identified by constraining four weight coefficients for each of four latent constructs to be equal one. (3) Parameter Estimation: its aim is to estimate population parameters by minimizing the difference between the observed and the implied model [87]. The maximum likelihood method, a widely used method, has been chosen as the estimation method in pursuit of the parameter values that provide the greatest benefit to the observed data. (4) Construct Validity: it examined the degree to which the proposed model fits the data [87]. To attain construct validity, several model fit indices should achieve their respective fitness thresholds. (5) Model Re-specification is concerned with improving the model fit by applying modification. Any decision regarding the model modification must be theoretically defensible [87]. After applying all the aforementioned steps to the theoretical model, the base model for the PCPS instrument has been created and then verified. For Study 1 and 2, the following model fits the indices, respectively, achieved: Chi-square/DF (1.96 and 2.06), CFI (0.94 and 0.94), GFI (0.91 and 0.92), RMSEA (0.062 and 0.061), and SRMR (0.050 and 0.052); where values of 5.0 and 3.0 are acceptable and good, respectively, for Chi-square/DF, values of 0.90 and 0.95 are acceptable and good, respectively, for CFI and GFI, and values of 0.08 and 0.06 are acceptable and good, respectively, for SRMR and RMSEA [90–93].

Appendix B.3.2. Model Comparison

After the construct validity (model fit) has been achieved, the last step of CFA (that is, model comparison) was performed. (6) Model comparison: it tests the sufficient number of factors (constructs) and respective observed variables for those factors (the structural model). If a scale were originally posited as containing multiple distinct factors (constructs), the measurement models should directly test this by comparing the fit of that model with more parsimonious nested models, including 1-factor, 2-factor, 3-factor models, etc. Two models are nested if one is derived from the other one by placing restrictions on it. Since the base model is originally a 4-factor model, all the best 3-factor, 2-factor, and 1-factor models derived from the base model were all nested to each other. (a) The best 3-factor model was nested with the new model and had one more constraint than the new model; the correlation between third and fourth factors constrained to be one (these two factors constrained to be totally dependent on each other). (b) The best 2-factor model was nested with the new model and had two more constraints than the base model, including the covariances among first, third, and fourth factors constrained to be one; i.e., all first, second, and third factors served as one single factor. The best 1-factor model was the original model in which all the covariances among four factors were constrained to be one. Chi-square difference test was conducted based on the below formulas shown in Equation (A1), and the results of these tests shown in Table A3:

$$\text{Chi-square difference test} = \chi^2 \text{ (model with fewer factors)} - \chi^2 \text{ (model with more factors)}/(\text{DF (fewer factor model)} - \text{DF (more factor model)}) \tag{A1}$$

The null and alternative hypothesis for all the following model comparisons using Chi-square difference test was:

$H_0$ comparison: There was no statistically significant difference between the base model (4-factor) and the fewer factor model, and the addition of the additional factor did not significantly improve the fit to the data; therefore, the base model is not preferred to the fewer factor model.

$H_1$ comparison: There was a statistically significant difference between the base model (4-factor) and the fewer factor model, and the addition of the additional factor did significantly improve the fit to the data; therefore, the base model is preferred to the fewer factor model.

According to Table A3, the statistical significance test for the difference between the base model and, respectively, 1-factor, the best 2-factor, and the best 3-factor models resulted in the rejection of the null hypotheses for both first and second studies. In other words, the deduction of the factors did not significantly improve the fit to the data; therefore, the

base model was preferred to the other rival nested models. This result emphasized that the sufficient number of factors for the PCPS instrument was four factors, which is the base model. The base model served as the final model for the PCPS instrument in measuring perception of CPS of individuals in the domain of complex systems.

**Table A3.** Comparisons of the base model with nested rival models.

| | Comparison between the Base Model and | $\Delta\chi^2$ | $\Delta DF$ | *p*-Value | Result | Decision |
|---|---|---|---|---|---|---|
| | The best 3-factor model | 82.8 | 1 | <0.001 | Reject $H_0$ | The base model selected |
| Study 1 | The best 2-factor model | 114.0 | 3 | <0.001 | Reject $H_0$ | The base model selected |
| | 1-factor model | 131.5 | 6 | <0.001 | Reject $H_0$ | The base model selected |
| | The best 3-factor model | 48.1 | 1 | <0.001 | Reject $H_0$ | The base model selected |
| Study 2 | The best 2-factor model | 68.3 | 3 | <0.001 | Reject $H_0$ | The base model selected |
| | 1-factor model | 103.8 | 6 | <0.001 | Reject $H_0$ | The base model selected |

*Appendix B.4. The Final Model*

After conducting the Chi-square difference test to verify the sufficient number of factors for the PCPS instrument, the base model was selected as the final model of the study. Table A4 shows the structure of the final model with respective factor loadings. The final model consisted of four distinct factors (constructs) and 17 items (questions), which measure different individual's PCPS. Validity and reliability features of the final model were demonstrated below:

**Table A4.** The final model of PCPS instrument after exploratory and confirmatory stages for practitioners and students.

| Factors | Item | Factor Loading (for Managers) | Factor Loading (for Students) |
|---|---|---|---|
| Problem Identification and Definition | Item 1 | 0.7 | 0.7 |
| | Item 2 | 0.6 | 0.5 |
| | Item 3 | 0.7 | 0.8 |
| | Item 4 | 0.5 | 0.6 |
| Information Gathering | Item 5 | 0.7 | 0.7 |
| | Item 6 | 0.6 | 0.7 |
| | Item 7 | 0.6 | 0.5 |
| | Item 8 | 0.2 | 0.5 |
| | Item 9 | 0.7 | 0.7 |
| | Item 10 | 0.9 | 0.8 |

**Table A4.** *Cont.*

| Factors | Item | Factor Loading (for Managers) | Factor Loading (for Students) |
|---|---|---|---|
| Evaluating Solutions and Developing Approaches | Item 11 | 0.7 | 0.8 |
| | Item 12 | 0.7 | 0.6 |
| | Item 13 | 0.6 | 0.6 |
| | Item 14 | 0.6 | 0.6 |
| Implementation Planning | Item 15 | 0.7 | 0.6 |
| | Item16 | 0.8 | 0.8 |
| | Item 17 | 0.8 | 0.9 |

1. Construct validity: For sample study 1 and sample study 2, the following model fits the indices, respectively, achieved: Chi-square/DF (1.96 and 2.06), CFI (0.94 and 0.94), GFI (0.91 and 0.92), RMSEA (0.062 and 0.061), and SRMR (0.050 and 0.052); where values of 5.0 and 3.0 are acceptable and good, respectively, for Chi-square/DF, values of 0.90 and 0.95 are acceptable and good, respectively, for CFI and GFI, and values of 0.08 and 0.06 are acceptable and good, respectively, for SRMR and RMSEA [90–93]. The construct validity's result suggested that the final model fitted the data well and was able to measure what was intended to measure.

2. Uni-dimensionality: This will be achieved when all measuring items have acceptable factor loadings for the related factor [85]. The sample size of this study was between 200 and 400, and according to Field [84] (pp. 440), factor loading greater than 0.4 on one factor demonstrates an acceptable relationship. As shown in Table A4, all the factor loading had acceptable and excellent factor loading. Therefore, the final model for both studies achieved the uni-dimensionality criterion.

3. Discriminant Validity: The covariance greater than 0.85 between two factors indicates the two factors are redundant or experiencing a serious multicollinearity problem [87]. Additionally, all the covariances between factors in the final model were below 0.85. Therefore, the final model had discriminant validity among its factors.

4. Composite Reliability (CR): Indicates the reliability and internal consistency of a latent factor (construct). The final model has achieved the CR criterion (CR > 0.7 and 0.8 are good and excellent, respectively) for all four factors (see Table A5) [94].

**Table A5.** Composite reliability results for the final model.

| Factors | Problem Identification and Definition | Information Gathering | Evaluating Solutions and Developing Approaches | Implementation Planning |
|---|---|---|---|---|
| Study 1 | 0.71 | 0.78 | 0.75 | 0.80 |
| Study 2 | 0.73 | 0.80 | 0.74 | 0.79 |

As has been discussed above, the final model respected all criteria of construct validity, uni-dimensionality, discriminant validity, and composite reliability. As a result, the main null hypothesis of the study ($H_0$main) was supported. There is no statistically significant difference between the final model of the PCPS instrument and the actual data model in order to measure the state of PCPS at the individual level; i.e., the final model of the PCPS instrument fits the data well and is able to measure the state of PCPS at the individual level.

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
