# Peer review of "Development of Perceived Complex Problem-Solving Instrument in Domain of Complex Systems"

_systems, doi:10.3390/systems9030051_

Round 1

Reviewer 1 Report

This paper reports the development of an instrument (PCPS) that purports to measure a style of "perception" conducive to solving complex problems, and a comparative study of its correlation with a previously developed measure (STSP) of problem solving skills.  The validation study is statistically strong, but the comparative study is weak as the same set of subjects were (apparently) used.  The overall presentation lacks clarity as detailed below.

The discussion is especially weak in its neglect of findings from the cognitive psychology of problem solving: only a single textbook is cited.  The authors should consider the substantial literature on variation in abstraction ability, symbolic problem solving ability, lateral thinking, analogy, and insight problem solving, including the relevant neuroscience.  These areas are now well-developed and are clearly relevant to the kinds of practical, industrial- and management-psychology kinds of issues being investigated here.

There are numerous unmatched ( or ) symbols, inappropriately captialized words, and incomplete or run-on sentences.  Several lists of items would be better as bullet lists.  English editing services are recommended.

Permission may be needed to reproduce Fig. 1 (from one of the current authors' previous work).

Specific comments:

ln. 73: There are three dimensions mentioned, not two.

ln. 241: Cognitive psychology is a massive field with 100s of papers published every month.  Was only a single textbook consulted?  See above.

ln. 292: The measure "alpha" (from ln. 335 presumably Cronbach's) should be explicitly defined, with a citation if appropriate.

ln. 295-311: The English version of the questionnaire described should be included as an appendix or online supplementary material.  The goal is for other groups to use your materials, with no changes, so that comparative studies are possible.

ln. 340: Were the members of the "panel of experts" not the authors?  If not, how were they identified and recruited?  Some sample of English and Persian versions should be included as an appendix so that readers fluent in both languages can make their own judgements about the translations.

ln. 356: Why were 108 student responses not analyzed?

Sect. 4 (ln 371 ff):  Which subjects participated in which stage of PCPS model development is unclear.  Were the "10 experts" (ln. 331) the subjects for the EFA/CFA stage, or were one or both of the larger sets of subjects used at this stage (as the caption of Table 5  and subsequent discussion suggests)?

ln. 382-516: This lengthy discussion of standard EFA/CFA should go into an appendix or supplementary material, not the main text.  Just summarize what was done in a paragraph.

Fig. 3 (ln 602): The eigenvalue labels are unreadable - put them above the values so you can use a larger font.  The captions needs to say which are professional and which are student values.  Put all of the definitions into the caption so that they are easier to read.  Are these values all experimental, or are some derived?  The variables \nu and \xi are never defined in the text - there should be an equation in the main text that defines these and the mapping between them, including the quality factor \beta.

ln. 604:  Here we go from validating the PCPS to comparing the PCPS and STSP, which is the main point of the study.  In principle, a completely separate group of subjects should have been used for this comparison, so that validation and comparison are not conflated.  This is a limited, preliminary study, so the same subjects are OK, but this possible confound should be noted explicitly and discussed.  For example, it is never clear which test was administered first.  The possibility that one test biases the results of the other on a single set of subjects needs to be discussed explicitly.

Reviewer 2 Report

An interesting paper aimed at developing a model to measure an individual's perception towards solving complex problems. The development and critical evaluation of the PCPS model (Perceived Complex Problem-Solving) are well structured and clearly defined. The data is collected through a questionnaire from two sources, practitioners and students. The practitioners are based in Iran, while the students are based in the USA – it is not clear if ‘culture’ influenced the data or if that would have made a difference anyhow. I am always sceptical of using students in research of this nature, but, in this case, it does give a different perspective to the study and possibly makes the model more universally applicable. It would have been helpful if a copy of the questionnaire (in English) was included as an appendix to the paper so the reader can judge the interpretation of the data. While ‘uncertainty’ is addressed in the model, one of the main issues in the perception of complexity in problem-solving is the perception of ‘risk’. Uncertainty and risk are not the same. The research framework is well laid out, and the factor analysis is sound.

There are many ‘typing errors’; for example, I refer the authors to line 43 (what is ‘still not clearly defined’ – is it the scholars? Should it be ‘is still not clearly defined? Which would then refer to ‘complex problem-solving), line 100 (the word ‘has’ is wrong), line 103 (‘It’ should be ‘it’) etc.…. The English grammar could be improved.

There are also errors in the references, for example. In [1 and 59], the titles of the papers are incorrect. There are some instances of ‘doi:http:/doi.org’. Either quote the DOI number or retrieved from (and date) http:doi.org….. I was also unable to locate some of the references, e.g., [49]. I found ref [46] on  https://doi.org/10.1002/sdr.198, but it states that it was first published on 26 January 2001. One of the references was accessed five years ago!

Round 2

Reviewer 1 Report

Thank you for these revisions.  There are now just a few typos to be fixed.